**Data Availability Statement:** Data has been uploaded to figshare: https://doi.org/10.6084/m9.figshare.21193693.

**Funding:** This work was funded by a grant from the Korea Health Technology R&D Project through the Korean Health Industry Development Institute

# Development of a gait speed estimation model for healthy older adults using a single inertial measurement unit

**Hyang Jun Lee**[1], **Ji Sun Park**[2], **Jong Bin Bae**[1], **Ji won Han**[1], **Ki Woong Kim**[1,2,3,4]*

**1** Department of Neuropsychiatry, Seoul National University Bundang Hospital, Seongnam, Korea, **2** Department of Brain and Cognitive Science, Seoul National University College of Natural Sciences, Seoul, Korea, **3** Department of Psychiatry, College of Medicine, Seoul National University, Seoul, Korea, **4** Department of Health Science and Technology, Seoul National University Graduate School of Convergence Science and Technology, Suwon, Korea

* kwkimmd@snu.ac.kr

## Abstract

Although gait speed changes are associated with various geriatric conditions, standard gait analysis systems, such as laboratory-based motion capture systems or instrumented walkways, are too expensive, spatially limited, and difficult to access. A wearable inertia sensor is cheap and easy to access; however, its accuracy in estimating gait speed is limited. In this study, we developed a model for accurately estimating the gait speed of healthy older adults using the data captured by an inertia sensor placed at their center of body mass (CoM). We enrolled 759 healthy older adults from two population-based cohort studies and asked them to walk on a 14 m long walkway thrice at comfortable paces with an inertia sensor attached to their CoM. In the middle of the walkway, we placed GAITRite™ to obtain the gold standard of gait speed. We then divided the participants into three subgroups using the normalized step length and developed a linear regression model for estimating the gold standard gait speed using age, foot length, and the features obtained from an inertia sensor, including cadence, vertical height displacement, yaw angle, and role angle of CoM. Our model exhibited excellent accuracy in estimating the gold standard gait speed (mean absolute error = 3.74%; root mean square error = 5.30 *cm/s*; intraclass correlation coefficient = 0.954). Our model may contribute to the early detection and monitoring of gait disorders and other geriatric conditions by making gait assessment easier, cheaper, and more ambulatory while remaining as accurate as other standard gait analysis systems.

## Introduction

Gait speed is associated with numerous geriatric conditions such as frailty, disability, falls, cognitive decline, and mortality [1–5]. However, standard gait analysis systems, such as laboratory-based motion capture systems or instrumented walkways, are too expensive, spatially limited, and difficult to access for simply estimating gait speed in older adults. Although gait speed can also be manually measured using a stopwatch, it is subject to human error and has

(KHIDI), funded by the Ministry of Health & Welfare, Republic of Korea (grant no. HI15C3206); , a grant from the Korean Health Technology R&D Project, Ministry of Health and Welfare, Republic of Korea (grant no. HI09C1379 [A092077], http://www.mw.go.kr) and a grant from the PlanB4U Co., Ltd. (Grant No. 06-2021-0273).

**Competing interests:** No authors have competing interests.

limited precision [5]. To overcome these limitations, several studies have attempted to estimate gait speed using wearable inertia sensors, which are far cheaper, less spatially constrained, easier to access than motion capture systems and instrumented walkways, and less susceptible to human error than stopwatches [6]. However, the accuracy of gait speed estimation using a wearable inertia sensor is still moderate because a wearable inertia sensor does not directly measure gait speed but estimates it indirectly through direct integration of acceleration [7], kinematic modeling and correction of gait motion [8], regression modeling of data [9, 10] or hybrid methods [11, 12]. Although we previously reported a regression model for estimating gait speed using a wearable inertia sensor in healthy older adults [10], our model had limited accuracy in case of older adults whose gait speed was slower than 100 cm/s [10].

Gait speed is the product of cadence and step length. Because we have demonstrated that cadence can be reliably and validly measured by a wearable inertia sensor over a wide range of gait speeds [13], the limited accuracy of gait speed estimated by a wearable inertia sensor may be attributed to the limited accuracy in estimating the gait features associated with step length. Step length is the product of motions in torso, knees, and feet, and is associated with the body weight that contributes to thrust power while walking [14, 15]. Therefore, we can improve the accuracy of gait speed estimation using a wearable inertia sensor by including the features associated with step length acquired from an inertia sensor, such as vertical height displacement, roll angle, yaw angle of the center of body mass (CoM), and body weight in the regression model for estimating gait speed. In addition, we can further improve the accuracy of gait speed estimation by stratifying the participants into sequential subgroups of step length and developing a regression model within each subgroup because a linear regression model is vulnerable to the degree of homogeneity of a sample [16] and the step length may change considerably with the health state in older adults [17].

In the current study, we developed step length subgroup-specific regression models for estimating gait speed in healthy older adults using age, foot length, and the features associated with step length acquired from a wearable inertia sensor placed at the CoM and examined their accuracy using the gait speed estimated by an instrumented walkway system.

## Methods

### Participants

As summarized in Table 1, we enrolled 1,058 participants from two cohort studies: 759 from the Korean Longitudinal Study on Cognitive Aging and Dementia (KLOSCAD) [18] and 299 from the Korean Frailty and Aging Cohort Study (KFACS) [19]. The KLOSCAD and KFACS are population-based prospective cohort studies of elderly Koreans. In the KLOSCAD, 6,818 Koreans aged 60 years and over who were randomly sampled from 13 districts across South Korea have been followed every two years since 2009. In the KFACS, 3,000 Korean volunteers aged 70–84 years were followed up every two years from 2016 to 2020. Among the 1,058 participants, 759 community-dwelling healthy older adults (338 men aged 74.8±5.0 years old and 421 women aged 73.3±4.5) were included in the current analysis after excluding participants with major psychiatric disorders, including mood disorders and neurocognitive disorders, neurologic disorders including Parkinson's disease and stroke, and musculoskeletal diseases that may affect gait or balance at the baseline assessment or any follow-up assessment. We also excluded participants whose Tinetti Performance Oriented Mobility Assessment (POMA) score was below 25 [20].

All the participants provided written informed consent themselves or via their legal guardians. This study was approved by the Institutional Review Board of the Seoul National University Bundang Hospital (IRB: B-2107-696-115).

**Table 1. Characteristics of the participants.**

| | All (N = 759) | Cohort | | | Sex | | |
|---|---|---|---|---|---|---|---|
| | | KLOSCAD (N = 532) | KFACS (N = 227) | p | Men (N = 338) | Women (N = 421) | p |
| Women (%) | 55.5 | 53.4 | 60.4 | .125 | | | |
| Age (year) | 73.9 ± 4.8 | 73.2 ± 5.1 | 75.6 ± 3.4 | <0.001 | 74.8 ± 5.0 | 73.3 ± 4.5 | <0.001 |
| MMSE (point) | 27.4 ± 2.3 | 27.6 ± 2.3 | 26.8 ± 2.1 | <0.001 | 27.5 ± 2.3 | 27.3 ± 2.3 | 0.160 |
| POMA (point) | 27.7 ± 0.6 | 27.7 ± 0.6 | 27.8 ± 0.6 | 0.748 | 27.8 ± 0.6 | 27.7 ± 0.6 | 0.449 |
| Height (cm) | 159.8 ± 8.1 | 160.2 ± 8.1 | 159.0 ± 8.1 | 0.155 | 166.5 ± 5.8 | 154.4 ± 5.1 | <0.001 |
| Weight (kg) | 61.6 ± 9.0 | 61.7 ± 9.0 | 61.2 ± 9.1 | 0.935 | 66.9 ± 8.3 | 57.3 ± 7.0 | <0.001 |
| BMI | 24.1 ± 2.6 | 24.0 ± 2.6 | 24.2 ± 2.7 | 0.239 | 24.1 ± 2.7 | 24.0 ± 2.6 | 0.896 |
| Overweight (%)* | 34.7 | 34.8 | 34.4 | .987 | 36.4 | 33.3 | .665 |
| Underweight (%)[†] | 1.8 | 1.9 | 1.8 | | 1.8 | 1.9 | |
| Foot length (cm) [‡], [¶] | 23.4 ± 1.4 | 23.4 ± 1.4 | 23.4 ± 1.4 | 0.949 | 24.5 ± 1.0 | 22.6 ± 0.9 | <0.001 |
| VHD (cm) [§], [¶¶] | 3.29 ± 0.77 | 3.27 ± 0.75 | 3.36 ± 0.83 | 0.034 | 3.55 ± 0.83 | 3.08 ± 0.65 | <0.001 |
| Cadence (steps/min) [¶] | 115.5 ± 9.4 | 114.8 ± 9.8 | 117.2 ± 8.3 | 0.001 | 113.0 ± 8.7 | 117.6 ± 9.5 | <0.001 |
| Gait speed (cm/s) [¶] | 114.4 ± 17.9 | 113.3 ± 18.7 | 117.0 ± 15.7 | 0.002 | 116.0 ± 18.4 | 113.1 ± 17.4 | 0.056 |
| Step length (cm) [¶] | 58.6 ± 7.0 | 58.4 ± 7.0 | 59.3 ± 7.1 | 0.037 | 60.8±7.3 | 56.9±6.2 | <0.001 |
| Roll angle ( ° )[¶¶] | 6.4 ± 2.4 | 6.1 ± 2.4 | 7.0 ± 2.4 | <0.001 | 5.3 ± 1.9 | 7.2 ± 2.5 | <0.001 |
| Yaw angle ( ° )[¶¶] | 12.2 ± 3.6 | 12.2 ± 3.7 | 12.3 ± 3.4 | 0.762 | 12.6 ± 3.7 | 12.0 ± 3.5 | 0.046 |

KLOSCAD, Korean Longitudinal Study on Cognitive Aging and Dementia; KFACS, Korean Frailty and Aging Cohort Study; MMSE, Mini Mental Status Examination; POMA, Tinetti Performance Oriented Mobility Assessment; VHD, vertical height displacement; BMI, body mass index

All values, except the percentages of women, overweight participants, and underweight participants, are presented as mean ± standard deviation.

*BMI ≥ 25, [†]BMI < 18.5, [‡]mean length of both feet

[§] Mean difference between the maximum and minimum vertical height of the center of mass within a step

[¶]measured using the GAITRite™

[¶¶] Estimated using an inertia measurement unit

## Clinical assessment

Geriatric psychiatrists or neurologists with expertise in dementia research performed face-to-face, standardized diagnostic interviews; physical and neurological examinations; and laboratory tests, including complete blood counts, chemistry profiles, serological tests for syphilis, echocardiography, and chest radiography. Research neuropsychologists or trained research nurses administered the CERAD-K neuropsychological assessment battery (CERAD-K-N), digit span test (DST), and frontal assessment battery (FAB) [21, 22].

We diagnosed dementia and other psychiatric disorders according to the DSM-IV [23] diagnostic criteria and determined the global severity of cognitive disorders using the clinical dementia rating (CDR) [24]. We evaluated gait and balance using the POMA. A higher POMA score represents better gait and balance, with a maximum score of 28 [20].

## Gait assessments

We measured the gait of each participant using an IMU (FITMETER® [FitLife Inc., Suwon, Korea] or ActiGraph® [SMD solution, Seoul, Korea]) placed over the center of body mass (CoM) and the GAITRite™ (CIR Systems Inc., Havertown, PA) simultaneously. The IMUs were hexahedrons (35 × 35 × 13 mm [14 g]/30 × 40 × 10 mm [17 g]) with smooth edges and a digital tri-axial accelerometer (BMA255, BOSCH, Germany) and gyroscope (BMX055, BOSCH, Germany). They could measure tri-axial acceleration up to ± 8 g (with a resolution of

0.004 g) and tri-axial angular velocity up to ±1,000˚/s (with a resolution of 0.03˚/s) at 250 Hz. We fixed an IMU to each participant at the 3rd– 4th lumbar vertebrae using Hypafix, a soft, stretchable, non-woven polyester material that adapts well to body contours. We asked each participant to walk back and forth three times on a 14 m flat straight walkway at a comfortable self-selected pace, and to start turning after passing the 14 m line. We placed the GAITRite electronic mat in the middle of the walkway to measure steady-state walking.

The GAITRite is a portable gait analysis walkway system connected to the USB port of a computer that measures temporal and spatial gait parameters via an electronic walkway at 100 Hz. Its walkway size is 520 (L) × 90 (W) × 0.6 cm (H), and it has an active sensing area of 427 (L) × 61 cm (W). It contains 16,128 sensors placed with a spatial accuracy of 1.27 cm.

## Development of the gait speed estimation models

To measure steady-state walking, we analyzed the data of the central 10m of 14m flat straight walkways, after eliminating the 2m-long walk prior to the start and each turn.

As illustrated in Fig 1, we preprocessed the IMU signals, selected features, and identified each step, as described in detail in our previous work [10]. We obtained the comparative gold standard gait speed from the GAITRite™. We then estimated the gait speed using IMU signals according to our previous regression model (Model 0) [10]. Model 0 included age, sex, sole length, cadence, and VHD. We then developed a new model (Model 1) by adding the roll and yaw angles at the CoM and body weight to Model 0. Step length is the product of motions in the torso, knees, and feet. The roll and yaw angles of the CoM reflect the angular motion of the torso and peripherals. The roll and yaw angles of the CoM were calculated and re-oriented to Cartesian coordinates from the sensor data from the IMU (Fig 1). Step length is also associated with body weight, which is related to the thrust power [14, 15].

Then, we divided the participants into three subgroups by k-means clustering of the normalized step length (NSL) and developed the subgroup-specific regression models (Model 2) with the features included in the Model 1. We estimated the step length by dividing the gait

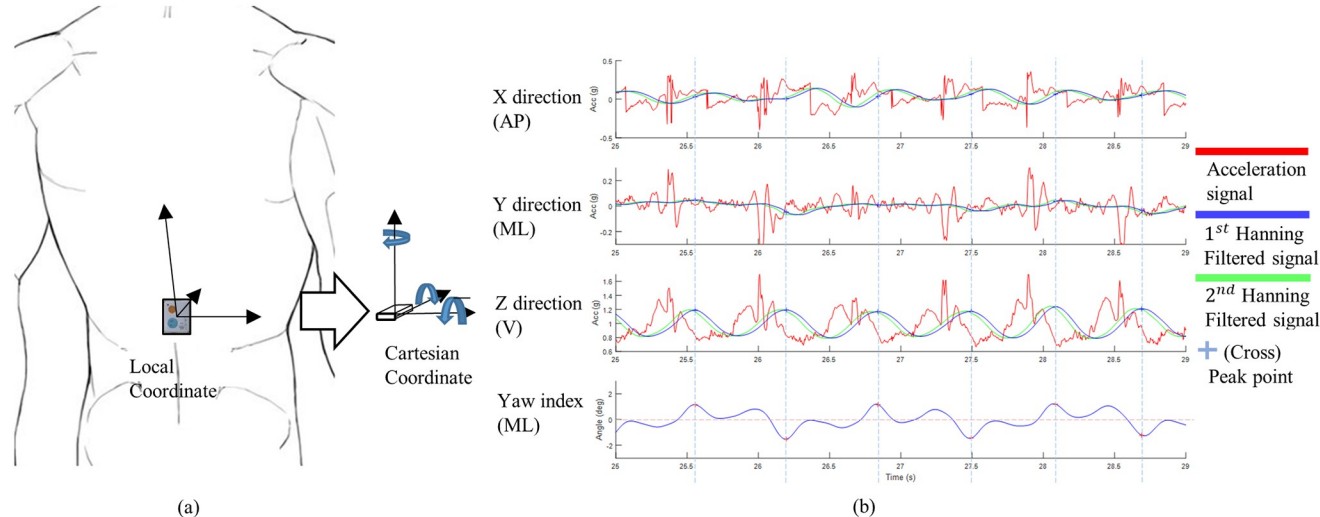

**Fig 1. Acquisition and preprocessing of the signals from a wearable inertia sensor.** (a) Three-dimensional acceleration signals were acquired from an inertia sensor placed at the center of body mass and re-oriented into Cartesian coordinates to correct the angular misplacement. (b) Hanning filter was applied to the raw acceleration signals and then each step was identified (indicated by crosses). The sides of steps (left or right) were determined by the yaw index at each step.

**Table 2. Characteristics of the subgroups classified by normalized step length.**

| | Short NSL (N = 188)[a] | Medium NSL (N = 378)[b] | Long NSL (N = 193)[c] | Statistics* | | |
| --- | --- | --- | --- | --- | --- | --- |
| | | | | F | P | posthoc |
| NSL | .324±0.019 | .370±.014 | .418 ±.019 | 1362 | < .001 | a < b < c |
| Women (%) | 51.4 | 57.2 | 56.6 | .430 | .651 | |
| Age (year) | 76.7 ± 5.1 | 73.2 ± 4.3 | 72.6 ± 4.3 | 37.9 | < .001 | a > b, c |
| Height (cm) | 159.7 ± 7.9 | 160.4 ± 8.2 | 158.8 ± 8.2 | 3.09 | .046 | b > c |
| Weight (Kg) | 62.3 ± 8.6 | 62.0 ± 9.0 | 60.1 ± 9.3 | 5.10 | .006 | a, b > c |
| Foot length (cm)[†] | 23.0 ± 1.2 | 23.5 ± 1.4 | 23.7 ± 1.4 | 14.0 | < .001 | a < b, c |
| Gait speed(cm/s)[†] | 95.1 ± 13.7 | 115.5 ± 12.2 | 130.9 ± 12.3 | 371 | < .001 | a < b < c |
| Cadence (steps/min) [‡] | 109.9 ± 9.8 | 116.4 ± 8.2 | 119.2 ± 8.7 | 49.8 | < .001 | a < b < c |
| VHD (cm) [‡], [§] | 2.53 ± 0.40 | 3.24 ± 0.47 | 4.14 ±0.69 | 408 | < .001 | a < b < c |
| Roll angle ( ° ) [‡],[¶] | 4.92 ± 1.67 | 6.42 ± 2.28 | 7.79 ± 2.53 | 71.5 | < .001 | a < b < c |
| Yaw angle ( ° ) [‡],[¶] | 9.56 ± 2.62 | 12.07 ± 2.93 | 15.15 ± 3.46 | 154 | < .001 | a < b < c |
| POMA (point) | 27.5 ± 0.8 | 27.8 ± 0.6 | 27.9 ± 0.4 | 18.8 | < .001 | a < b, c |

NSL, normalized step length grouped by k-means clustering; VHD, vertical height displacement (mean of the maximum and minimum vertical heights within a step);
POMA, Tinetti Performance Oriented Mobility Assessment

All values except the percentages for women are presented as mean ± standard deviation.

* Analysis of variance with Bonferroni post-hoc comparison

[†]measured by the GAITRite™;

[‡]estimated by an inertia measurement unit.

[§] Mean of the maximum and minimum vertical heights within a step.

[¶] Difference between the minimum and maximum yaw angles within a stride

speed obtained from the Model 1 by the cadence and calculated the NSL by dividing the step length by the height. The characteristics of the three NSL subgroups are summarized in Table 2. For all the regression models, only the features with a variation inflation factor (VIF) below 2.5 were included.

## Statistical analyses

Continuous variables were compared using a student's t-test or analysis of variance (ANOVA) and categorical variables were compared using the chi-square test between groups. We calculated the intraclass correlation coefficient (ICC), mean error (ME), mean absolute error (MAE), and root mean square error (RMSE) of the gait speed estimated from the gait speed estimation models and compared them to those measured by the Gaitrite™. We then compared the ICC, ME, MAE, and RMSE of the estimated gait speeds obtained with the gait speed estimation models using repeated measures ANOVA. All the statistical analyses were performed using the Statistical Package for the Social Sciences version 25.0 (International Business Machines Corporation, Armonk, NY).

## Results

As summarized in Table 3, Model 2 included all features of Model 1, roll angle, yaw angle, and body weight, except the feature in Model 0, sex of the participants. Although Model 0 estimated the gait speed measured by the Gaitrite™ quite well (R = 0.935, $R^2$ = 0.875, adjusted $R^2$ = 0.874, F = 1051, p < 0.001), Model 1 estimated it better (R = 0.953, $R^2$ = 0.908, adjusted $R^2$ = 0.908, F = 1064, p < 0.001). When we developed the NSL subgroup-specific gait speed

**Table 3. Development of the gait speed estimation model by employing the body weight and roll and yaw angles of the center of mass as additional predicting features.**

| | Model 0 [10]* | | | | Model 1 | | | |
|---|---|---|---|---|---|---|---|---|
| | β (SE) | B | p | VIF | β (SE) | B | p | VIF |
| Constant | -113.2 (8.61) | | < .001 | | -106.0(6.13) | | < .001 | |
| Age (year) | -.388(.050) | -.104 | < .001 | 1.09 | -.328(.043) | -.088 | < .001 | 1.10 |
| Sex | 3.06(.676) | .085 | < .001 | 2.12 | | | | |
| Cadence (steps/min) † | 1.17(.026) | .617 | < .001 | 1.14 | 1.10(.023) | .578 | < .001 | 1.18 |
| VHD (cm) † | 12.1(.339) | .521 | < .001 | 1.28 | 10.1(.312) | .436 | < .001 | 1.48 |
| Foot length (cm)‡,§- | 3.25(.246) | .250 | < .001 | 2.15 | 3.29(.208) | .253 | < .001 | 2.10 |
| Weight (Kg) | | | | | -0.115(.029) | -.058 | < .001 | 1.69 |
| Roll angle( ° )†,¶ | | | | | 1.01(.089) | .137 | < .001 | 1.20 |
| Yaw angle( ° )†,¶ | | | | | 0.647(.064) | .130 | < .001 | 1.37 |

VIF, variation inflation factor; VHD, vertical height displacement (mean of the maximum and minimum vertical heights within a step).

Sex: coded (male = 1, female = 2)

* Gait speed estimation model developed in our previous study that did not include body weight and the yaw angle of the center of mass as predicting features

†estimated using an inertia measurement unit

‡measured using the GAITRite™

§ Mean length of both feet

estimation models using the features included in Model 1, all the features were selected as significant predictors of gait speed in all the NSL subgroups with low VIF (Table 4). This NSL subgroup-specific model (Model 2) estimates gait speed better than Model 0 and Model 1 ($R = 0.955$, $R^2 = 0.912$, adjusted $R^2 = 0.912$, $p < 0.001$). In all the NSL subgroups, the gait speed estimates were satisfactory (adjusted $R^2 = 0.828$, $F = 129.545$, $p < 0.001$ for the short NSL subgroup; adjusted $R^2 = 0.801$, $F = 217.135$, $p < 0.001$ for the medium NSL subgroup; adjusted $R^2 = 0.836$, $F = 140.971$, $p < 0.001$ for the long NSL subgroup).

**Table 4. Development of the normalized step length subgroup-specific gait speed estimation models.**

| | Short NSL (N = 188) | | | | Medium NSL (N = 378) | | | | Long NSL (N = 193) | | | |
|---|---|---|---|---|---|---|---|---|---|---|---|---|
| | β (SE) | B | p | VIF | β (SE) | B | p | VIF | β (SE) | B | p | VIF |
| Constant | -95.6(12.6) | | < .001 | | -117.8(8.79) | | < .001 | | -121.4(13.3) | | < .001 | |
| Age | -.462(.086) | -.172 | < .001 | 1.11 | -.327(.067) | -.116 | < .001 | 1.58 | -.272(.086) | -.095 | .002 | 1.06 |
| Cadence* | 1.06 (.043) | .760 | < .001 | 1.06 | 1.14(.035) | .773 | < .001 | 1.08 | 1.15(.050) | .817 | < .001 | 1.47 |
| VHD*,† | 13.3(1.10) | .388 | < .001 | 1.12 | 11.8(.692) | .455 | < .001 | 1.36 | 9.90(.622) | .555 | < .001 | 1.42 |
| Foot length‡ | 3.24(.451) | .287 | < .001 | 1.74 | 3.26(.301) | .372 | < .001 | 2.25 | 3.68(.386) | .424 | < .001 | 2.31 |
| Weight | -.154(.063) | -.096 | .016 | 1.71 | -0.122(.044) | -.090 | .005 | 1.98 | -.179(.050) | -.136 | < .001 | 1.70 |
| Roll angle*,§ | 1.27(.264) | .155 | < .001 | 1.13 | 1.14(.134) | .212 | < .001 | 1.19 | 1.04(.154) | .213 | < .001 | 1.18 |
| Yaw angle*,§ | .550(.164) | .105 | .001 | 1.07 | .828(.097) | .199 | < .001 | 1.04 | .620(.109) | .174 | < .001 | 1.10 |

NSL, normalized step length grouped by k-means clustering; VHD, vertical height displacement (mean of the maximum and minimum vertical heights within a step)

*estimated using an inertia measurement unit

† Mean of maximum and minimum vertical heights within a step

‡ Mean of the lengths of both feet measured using the GAITRite™

§ Difference between the minimum and maximum yaw angles within a stride

**Table 5. Comparison of the accuracies of the three gait speed estimation models.** The values predicted by the models were compared to the values obtained with the GAITRite™.

| | Slow (N = 110)* | Medium (N = 536)* | Fast (N = 113) * | All (N = 759) |
|---|---|---|---|---|
| Gait speed | | | | |
| GAITRite[a] | 85.2 ± 9.2 | 114.6 ± 9.4 | 141.6 ± 8.9 | 114.4 ± 17.9 |
| Model 0[b] | 88.2 ± 10.4 | 114.8 ± 10.1 | 137.8 ± 9.2 | 114.4 ± 16.8 |
| Model 1[c] | 87.6 ± 10.3 | 114.7 ± 10.0 | 138.9 ± 8.8 | 114.4 ± 17.1 |
| Model 2[d] | 87.5 ± 10.4 | 114.7 ± 10.1 | 138.9 ± 8.6 | 114.4 ± 17.1 |
| p[†] | <0.001 | 0.728 | <0.001 | 1.00 |
| posthoc[†] | a < b, c, d | - | a > c, d > b | |
| ME | | | | |
| Model 0[a] | 3.63 | .268 | -2.64 | .323 |
| Model 1[b] | 2.92 | .127 | -1.85 | .237 |
| Model 2[c] | 2.75 | .144 | -1.85 | .225 |
| p[†] | .047 | .462 | .009 | .691 |
| posthoc[†] | a > c | - | a < b, c | |
| MAE | | | | |
| Model 0[a] | 6.05 | 4.08 | 4.20 | 4.38 |
| Model 1[b] | 5.68 | 3.57 | 3.32 | 3.84 |
| Model 2[c] | 5.37 | 3.52 | 3.18 | 3.74 |
| p[†] | .043 | < .001 | < .001 | < .001 |
| posthoc[†] | | a > b, c | a > b, c | a > b > c |
| RMSE | | | | |
| Model 0 | 6.86 | 5.89 | 7.71 | 6.34 |
| Model 1 | 6.16 | 5.03 | 6.34 | 5.42 |
| Model 2 | 6.01 | 4.99 | 5.96 | 5.30 |
| ICC | | | | |
| Model 0 | .766 ± .088[‡] | .817 ± .027[‡] | .665 ± .133[‡] | .933 ± .009[‡] |
| Model 1 | .806 ± .070[‡] | .866 ± .020[‡] | .753 ± .092[‡] | .952 ± .006[‡] |
| Model 2 | .817 ± .065[‡] | .869 ± .019[‡] | .778 ± .089[‡] | .954 ± .006[‡] |

IMU: inertia measurement unit; ME: mean error (%); MAE: mean absolute error (%); RMSE: root mean square error (cm/s); ICC: intraclass correlation coefficient.

* Gait speed measured by the GAITRite™.

The medium speed was defined as the gait speed within one standard deviation from the average (114.37 m/s±17.92). The slow and fast speeds were defined as slower and faster than the medium speed, respectively.

[†] rmANOVA with Bonferroni post-hoc comparisons.

[‡] $p < 0.001$ by ICC [1, 3]

We then compared the accuracies of the gait speed estimations of the models by employing the gait speed measured by the GAITRite™ as a comparative standard (Table 5). For all participants, Models 1 and 2 showed lower MAE, ME, and RMSE and better ICC than Model 0. Model 2 showed a lower MAE than Model 1 (p = 0.007). We then grouped the participants into three subgroups (fast, intermediate, and slow) using the gait speed measured by the GAITRite™. Slow gait was defined as one standard deviation below the average (17.9 m/s) and fast gait as one standard deviation above the average (114.4 m/s). The gait speed estimated by Model 0 tended to be slower than that measured by the GAITRite™ in the fast gait speed subgroup (p < 0.001) and faster in the slow gait speed subgroup (p < 0.001). Although this trend persisted in the gait speeds estimated by Model1 and Model 2, the MAE in the gait speed

estimates obtained with an IMU and GAITRite™ were significantly reduced for the fast and medium gait subgroups (p < 0.001).

## Discussion

Gait speed estimation by linear regression is essentially a method for approximating the true mathematical expression of human gait speed. A linear regression is a simplified function of Taylor expansion form of the true mathematical expression of gait which may include higher degree functions and number of variables. For usability, the features selected in a linear regression should be obtainable, and the Taylor $1^{st}$ order local regression is a method for approximation. This study demonstrated that the accuracy of gait speed estimation using an IMU can be significantly improved by employing features associated with step length in the linear regression model and optimizing the linear regression model in the NSL subgroups.

The three additional features selected in the new linear regression models were body weight and the roll and the yaw angles of the CoM. The effect of body weight on gait speed is complex [25]. Both weight loss and gain may reduce gait speed by reducing muscle power [15, 26, 27] and/or inducing physical frailty [28]. However, in our older adult sample, a BMI of 25 or higher, which is considered overweight, was more common than a BMI < 18.5, which is considered underweight, in both sexes (Table 1). This may be the reason why body weight was selected in the models and why their coefficients were negative in the regression.

Body rotation was related to walking pattern. The yaw angle represents the three-dimensional walking pattern. From the top view, the yaw rotation adds the CoM movement to the forward ground motion of the stance leg when subjects rotate the waist in steps by each leg moving forward [12]. In our models, the coefficient of the yaw angle was positive, indicating that the roll angle may increase as the gait speed increases. However, there have been no previous studies on the relationship between the yaw angle of the CoM and gait speed. The roll angle represents the movement related to mediolateral (ML) stability and is related to gait speed. In our models, the coefficient of the roll angle was positive, indicating that the roll angle may increase as the gait speed increases. This result is in line with that of a previous study. Lee et al. also found that the roll angle increased as gait speed increased in normal older adults [29].

Sex, which was included in our previous model (Model 0), was not selected in our new models (Model 1 and Model 2). Among the three additional features in our new models, the roll angle of CoM, which was selected in the new models, might have nullified the effect of sex on gait speed. As shown in Table 1, women had a larger roll angle than men. However, the yaw angle was not statistically different between the sexes.

The step length can be changed by body kinematics and muscle power [30–32], and reduces with advancing age [15, 26]. Older adults may have various aging-associated problems to some degree that may change their gait [33]. To maintain gait stability, older adults with musculoskeletal degenerative changes may have shorter step lengths than those without musculoskeletal degenerative changes [34]. Because gait speed is also influenced by step width and body weight [25, 35] we employed the NSL to develop subgroup-specific models for gait speed estimation in the current study. Human gait in the stance phase can be modelled as a spring-loaded inverted pendulum (Fig 1) [36, 37]. The stance leg length is expressed as

$$l_l = \beta_1(t)l_1$$

where $l_l$ is the leg length, and $\beta_1$ is the coefficient of leg bending.

Considering the symmetry of motion, the step length and NSL can be calculated from the kinematics as follows:

$$l_s \approx 2l_1\beta_1(0)sin(a)$$

$$\text{NSL} = \frac{l_s}{h} \approx \frac{2l_1}{h}\beta_1(0)sin(a) = 2\alpha_1\beta_1(0)sin(a)$$

where $m_1$ is the point mass equivalent to the body, $\omega_1$ is the angular velocity before contact, $l_l$ is the leg length, $l_s$ is the step length, $\beta_1(0)$ is the leg-bending coefficient at the contact point, $\alpha$ is the angle between the span leg and vertical line, which is assumed to be symmetric, and $\alpha_i$ is the human leg/height ratio. The step length is related to walking efficiency and external forces [38]. Considering the conservation of angular momentum and point mass assumption, the energy conserved after contact is as follows:

$$E_{con} = \frac{1}{2}m_1l_l^2\omega_1^2\left[1 - \left(\frac{l_s^2}{4l_1^2}\right)^2\right]$$

From the conserved energy equation, the step length $l_s$ is critical for energy conservation between steps. As the step length increases, the conserved energy decreases, which means that a larger external energy/force is needed to maintain the gait speed and step length, which is more dynamic and faster than that at small step lengths.

As shown in Table 4, the three subgroups may have different gait patterns associated with different anthropometric characteristics. In Model 2, the coefficient of VHD was highest in the short NSL subgroup and lowest in the long NSL subgroup, while B was smallest in the short NSL subgroup, indicating that the short NSL subgroup had less dynamic gait and less effective body rotation than the other NSL subgroups. In addition, the coefficient and B value of age were lowest in the long NSL subgroup, suggesting that the gait of the long NSL subgroup may be less affected by age.

This study has several limitations. First, all participants were healthy older adults. Therefore, the performance of our gait speed estimation algorithm may be reduced in frail older adults, who may have quite different gait patterns from healthy older adults. Second, a regression analysis was applied to the dataset. Although we divided the dataset into subgroups, the subgroup-specific linear models may be subject to underfitting errors because of model simplicity. Third, we used the signals from a single IMU. Although the IMU was placed at the CoM, it may not be sufficient to capture the gait characteristics associated with gait speed. Multiple sensors, such as sensor network systems, can improve the accuracy of gait estimation [39–41]. Fourth, although we fixed an IMU to the CoM using Hypafix and reoriented the signals from the IMUs before developing our models, it is difficult to completely rule out the possibility that there is a modest difference in the position and/or angle of the IMU between the participants because of inter-individual differences in body shape and human errors.

Despite these limitations, our model may contribute to improving the screening and monitoring of various geriatric conditions in older adults. Most smartphone models have recently been equipped with inertia sensors. Therefore, without a separate external inertia sensor, the smartphone itself can be used simultaneously as a sensor as well as a signal processing device. This makes it possible to use our model easily, cheaply, and widely.

## Author Contributions

**Conceptualization:** Hyang Jun Lee, Ki Woong Kim.

**Data curation:** Hyang Jun Lee, Ji Sun Park, Jong Bin Bae, Ji won Han.

**Formal analysis:** Hyang Jun Lee, Ki Woong Kim.

**Methodology:** Hyang Jun Lee, Ki Woong Kim.

**Project administration:** Jong Bin Bae, Ji won Han, Ki Woong Kim.

**Software:** Hyang Jun Lee.

**Supervision:** Jong Bin Bae, Ji won Han, Ki Woong Kim.

**Writing – original draft:** Hyang Jun Lee.

**Writing – review & editing:** Hyang Jun Lee, Ki Woong Kim.

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
