## [Decision Letter · Decision Letter 0]

21 Jul 2022

PONE-D-22-00890Development of a gait speed estimation model for healthy older adults using a single inertial measurement unitPLOS ONE

Dear Dr. Kim,

Thank you for submitting your manuscript to PLOS ONE. After careful consideration, we feel that it has merit but does not fully meet PLOS ONE’s publication criteria as it currently stands. Therefore, we invite you to submit a revised version of the manuscript that addresses the points raised during the review process.

We look forward to receiving your revised manuscript.

Kind regards,

Yaodong Gu

Academic Editor

PLOS ONE

Journal Requirements:

2. Please include a caption for figure 1.

Additional Editor Comments (if provided):

Discussion part shall be more detail to describe the gait analysis methods.

Reviewers' comments:

Reviewer's Responses to Questions

**Comments to the Author**

1. Is the manuscript technically sound, and do the data support the conclusions?

Reviewer #1: Yes

Reviewer #2: Yes

2. Has the statistical analysis been performed appropriately and rigorously? 

Reviewer #1: Yes

Reviewer #2: Yes

3. Have the authors made all data underlying the findings in their manuscript fully available?

Reviewer #1: Yes

Reviewer #2: Yes

4. Is the manuscript presented in an intelligible fashion and written in standard English?

Reviewer #1: Yes

Reviewer #2: Yes

5. Review Comments to the Author

Reviewer #1: This manuscript entitled “Development of a gait speed estimation model for healthy older adults using a single inertial measurement unit” primarily aimed to develop an algorithm for estimating the gait speed of healthy older adults using data captured by an inertial measurement unit placed at the center of body mass. To enhance the quality of the manuscript, revise suggestions are given below.

At the beginning of the whole study, the author needs to present your work as less word as you can, which include purpose, methods, results and conclusion. During the abstract part, it cannot catch the redear attention at the first time. Please rewrite this part.

Please further prove why this study is important. We have to confirm that this research is meaningful, rather than there is no such study that has been done yet. Author needs to well-introduce what kind of problems or issues would be solved after this research.

Please make your methods part more specific with some test images, and make your method more specific as well. This short explanation cannot easily understand.

Did the author perform Shapiro-Wilk normality tests?

Reviewer #2: This paper developed an algorithm for estimating the gait speed of healthy older adults using data captured by an inertial measurement unit (IMU) placed at the center of body mass (CoM). The model exhibited excellent accuracy in estimating the gait speed measured by the GAITRite. This quantitative gait evaluation method is completely derived from the information of IMUs. This technology has vast implications for gait estimation if properly validated. The work is interesting, however, there are a few changes that could be made to make the results more accessible and clear to readers, in details:

1.The abstract is not clear. The significance and challenges should be presented in the abstract.

2. The motivations should be strengthened when revising in the introduction. The authors should further enlarge Introduction with current results to improve the research background.

3. What are the limitations of the proposed gait estimation method? The author did not mention it in the paper.

4. It would be nice to discuss the following papers in the paper:

- Zhang, M., Wang, Q., Liu, D., Zhao, B., Tang, J., & Sun, J. Real-time gait phase recognition based on time domain features of multi-MEMS inertial sensors. IEEE Transactions on Instrumentation and Measurement, 2021,70, 1-12.

- Marta, G., Alessandra, P., Simona, F., Andrea, C., Dario, B., Stefano, S., Stefano, M.. Wearable Biofeedback Suit to Promote and Monitor Aquatic Exercises: A Feasibility Study. IEEE Transactions on Instrumentation and Measurement, 2020, 69(4), 1219-1231.

- S. Qiu, H. Zhao, N. Jiang, D. Wu, G. Song, H. Zhao, Z. Wang. Sensor network oriented human motion capture via wearable intelligent system, International Journal of Intelligent Systems, 2022, 37(2): 1646-1673.

5.Transitions from section to section should be smoother

6. What are the implications of the findings? More discussion should be provided in the manuscript. The factors that influence the accuracy of gait estimation should be analyzed in more detail in the discussion section.

7. How do you deal with inertial sensor misplacement?

8. Some individuals are not willing to wear additional sensors on their body, what is the potential of smartphone serving as the data collection tool?

9.Proofread the paper and improve readability.

6. PLOS authors have the option to publish the peer review history of their article (what does this mean?). If published, this will include your full peer review and any attached files.

Reviewer #1: No

Reviewer #2: No

---

## [Author Response · Author response to Decision Letter 0]

19 Aug 2022

Dear Editor,

Thank you for your comments on our work titled as "Development of a gait speed estimation model for healthy older adults using a single inertial measurement unit”. 

Our answers to the comments that the reviewers proposed are listed in the following pages. The revisions we made are indicated in the answers and highlighted in the manuscript. 

We appreciate it very much that you are considering publishing our paper in your journal.

.

Sincerely yours,

Ki Woong Kim, M.D., Ph.D.

 

Responses to the reviewers' comments

[Reviewer #1]

1. At the beginning of the whole study, the author needs to present your work as less word as you can, which include purpose, methods, results and conclusion. During the abstract part, it cannot catch the reader attention at the first time. Please rewrite this part.

(Answer)

Thank you for the comment. We totally agree with your comment and revised the abstract accordingly (PAGE 2 LINE 23-41). 

2. Please further prove why this study is important. We have to confirm that this research is meaningful, rather than there is no such study that has been done yet. Author needs to well-introduce what kind of problems or issues would be solved after this research.

(Answer)

Thank you for the comment. We extensively revised the introduction section accordingly (PAGE 3-4 LINE 46 - 78). 

3. Please make your methods part more specific with some test images and make your method more specific as well. This short explanation cannot easily understand.

(Answer)

Thank you for the comment. We did not describe the details of gait analysis in the method section because they are quite heavy and described in detail in our previous work cited in the current manuscript. However, for the better understanding of the readers, we added a brief summary on it with a figure (Figure 1) in the revised manuscript (PAGE 9-10 LINE 140-168 ) . 

4. Did the author perform Shapiro-Wilk normality tests?

(Answer)

Thank you for your comment. We performed the Shapiro-Wilk normality test on the gait speed acquired from the GAITriteTM. P value was 0.159 in men and 0.203 in women, indicating that the gait speed was normally distributed. 

[Reviewer #2]

1. The abstract is not clear. The significance and challenges should be presented in the abstract.

(Answer)

Thank you for the comment. We totally agree with your comment and revised the abstract accordingly (PAGE 2 LINE 23-41). 

2. The motivations should be strengthened when revising in the introduction. The authors should further enlarge Introduction with current results to improve the research background.

(Answer)

Thank you for the comment. We extensively revised the introduction section accordingly (PAGE 3-4 LINE 46 - 78). 

3. What are the limitations of the proposed gait estimation method? The author did not mention it in the paper.

(Answer)

Thank you for the comment. We described the limitations of our methods in the discussion section in more detail (PAGE 19 – 20 LINE 320-336 ). 

4. It would be nice to discuss the following papers in the paper:

- Zhang, M., Wang, Q., Liu, D., Zhao, B., Tang, J., & Sun, J. Real-time gait phase recognition based on time domain features of multi-MEMS inertial sensors. IEEE Transactions on Instrumentation and Measurement, 2021,70, 1-12.

- Marta, G., Alessandra, P., Simona, F., Andrea, C., Dario, B., Stefano, S., Stefano, M.. Wearable Biofeedback Suit to Promote and Monitor Aquatic Exercises: A Feasibility Study. IEEE Transactions on Instrumentation and Measurement, 2020, 69(4), 1219-1231.

- S. Qiu, H. Zhao, N. Jiang, D. Wu, G. Song, H. Zhao, Z. Wang. Sensor network oriented human motion capture via wearable intelligent system, International Journal of Intelligent Systems, 2022, 37(2): 1646-1673.

(Answer)

Thank you for the recommendation. The limitation of the gait speed estimation using a single sensor can be overcome by employing the multiple sensors and sensor network system. We point out this in the discussion section and cited the references you recommended. (PAGE 20 LINE 325-328) 

5. Transitions from section to section should be smoother

(Answer)

Thank you for the comment. We tried to revise the transitions smoother in the introduction, methods and discussion accordingly.

6. What are the implications of the findings? More discussion should be provided in the manuscript. The factors that influence the accuracy of gait estimation should be analyzed in more detail in the discussion section.

(Answer)

Thank you for the comment. We added the implications of our model in the discussion section (PAGE 20 LINE 332-336). We discussed the factors such as walking pattern and step length that improved the accuracy of gait speed estimation in our new models (PAGE 18-19 LINE 288-319). In addition, we added a brief discussion on the misplacement of the sensor which might have also influence the accuracy of gait speed estimation (PAGE 20 LINE 328-331). 

7. How do you deal with inertial sensor misplacement?

(Answer)

Thank you for the comment. As we described in the method section, we fixed an IMU to each participant at the 3rd – 4th lumbar vertebrae using Hypafix which is the soft, stretchable non-woven polyester material adapts well to body contours. We revised the method section in more detail and added the possibility of misplacement error as an additional limitation of our study (PAGE 8 LINE 129-130, PAGE 20 LINE 328-331).

8. Some individuals are not willing to wear additional sensors on their body, what is the potential of smartphone serving as the data collection tool?

(Answer)

Thank you for the comment. We added a discussion on the potential application of our model to a smartphone which can be employed as an inertia sensor as well as an acceleration signal processor. (PAGE 20 LINE 332-336).

9. Proofread the paper and improve readability.

(Answer)

Thank you for the comment. We got a proofread of the revised manuscript again from a professional proofreading service.

---

## [Decision Letter · Decision Letter 1]

20 Sep 2022

Development of a gait speed estimation model for healthy older adults using a single inertial measurement unit

PONE-D-22-00890R1

Dear Dr. Kim,

We’re pleased to inform you that your manuscript has been judged scientifically suitable for publication and will be formally accepted for publication once it meets all outstanding technical requirements.

Kind regards,

Yaodong Gu

Academic Editor

PLOS ONE

Additional Editor Comments (optional):

N/A

Reviewers' comments:

Reviewer's Responses to Questions

**Comments to the Author**

1. If the authors have adequately addressed your comments raised in a previous round of review and you feel that this manuscript is now acceptable for publication, you may indicate that here to bypass the “Comments to the Author” section, enter your conflict of interest statement in the “Confidential to Editor” section, and submit your "Accept" recommendation.

Reviewer #1: (No Response)

Reviewer #2: All comments have been addressed

2. Is the manuscript technically sound, and do the data support the conclusions?

Reviewer #1: (No Response)

Reviewer #2: Yes

3. Has the statistical analysis been performed appropriately and rigorously? 

Reviewer #1: (No Response)

Reviewer #2: Yes

4. Have the authors made all data underlying the findings in their manuscript fully available?

Reviewer #1: (No Response)

Reviewer #2: Yes

5. Is the manuscript presented in an intelligible fashion and written in standard English?

Reviewer #1: (No Response)

Reviewer #2: Yes

6. Review Comments to the Author

Reviewer #1: (No Response)

Reviewer #2: This paper proposed a model for accurately estimating the gait speed of healthy older adults using the data captured by an inertia sensor placed at their center of body mass (CoM). I value the authors efforts to answer all the previous concerns and feel that the paper overall improved. It seems to me that most of the previous concerns were well addressed in the revised manuscript. Overall, the revised paper quality largely meets the requirements of "PLOS ONE", hence the manuscript could be accepted for publication.

7. PLOS authors have the option to publish the peer review history of their article (what does this mean?). If published, this will include your full peer review and any attached files.

Reviewer #1: No

Reviewer #2: No

---

## [Editor Report · Acceptance letter]

27 Sep 2022

PONE-D-22-00890R1 

Development of a gait speed estimation model for healthy older adults using a single inertial measurement unit 

Dear Dr. Kim:

I'm pleased to inform you that your manuscript has been deemed suitable for publication in PLOS ONE. Congratulations! Your manuscript is now with our production department. 

Kind regards, 

on behalf of

Professor Yaodong Gu 

Academic Editor

PLOS ONE